# Applying spatial attention-based autoencoder learning of latent representation for unsupervised characterization of tumor microenvironment

## Abstract

Spatial tissue imaging technologies enable highly resolved spatial characterization of cellular phenotypes however today it depends on laborious manual annotation and molecular labels to understand tissue organization. As a result, we are not optimally leveraging higher-order patterns of cell organization potentially connected to disease pathology. Our approach combines information on cellular phenotypes with the physical proximity of cells to accurately identify organ-specific microanatomical in the tumor microenvironment.

**Keywords:** Autoencodeurs, spatial attention, computational pathology

## 1. Introduction

Spatial biology applications are advancing the study of tissue organization and cell2cell communication at an unprecedented scale. New tools are required to store, analyze and visualize the large diversity of produced data(Barmpoutis et al., 2021)(Ahmedt-Aristizabal et al., 2022). Autoencoders (AE) have been used to visualize or generate interpretable embeddings from biological single-cell data, an example is scvis(Ding et al., 2018)(Becht et al., 2019). More recently, new autoencoder models, in particular graph-based AE, use spatial information along with phenotypical information(Hu et al., 2021)(Dong and Zhang, 2022). But attention neural networks are becoming popular and new autoencoder models using attention mechanism have shown promising results with biological data(Dong and Zhang, 2022)(Fu et al., 2021). Imaging mass cytometry (IMC) it is a novel imaging technology, with more than 40 markers per pixel, that due the high content natures poses computational complexity challenges. Furthermore, due to the measurement protocol, the data can be noisy prone to artefacts and create biaises difficult to handle by the clustering approaches. To address all these challenges we are proposing a new approach to leverage spatial attention AE for IMC tissue structure phenotyping.

## 2. Materials and methods

**Data:** Patients included in the study are advanced stage and metastatic treated with different immune-oncology therapies(Camps et al., 2023). IMC assays were applied to selected ROIs with 40-marker panel (DAPI + 39 molecular probes) to characterize the TME. ROIs (3-4 per patient) were selected by an expert histologist to represent at high resolution (1 micro-meter) major tumor anatomical regions (Figure.1A). **Data split:** To fully leverage all available data (Table.1) we used dataset I (2 patients, 8 ROIs) for training and dataset II (39 patients, 128 ROIs) and III (139 patients, 556 ROIs) for independent testing.

**Design of autoencoder experimentations:** **(Classic AE)** The model considered as a baseline it is a regular fully connected three-layer autoencoder with a wasserstein regularization (loss term using Wasserstein distance on the embeddings)(Kolouri et al., 2018). **(Classic spatially aware AE)** For each cell, a value representing the distance to specified cell populations (annotated histological ROIs) is constructed. Outlier detections algorithms (i.e. Isolation Forest algorithm) is used to encode with min-max scaling, 1 (cell is close to a histological cluster) and 0 (it is not part a histological ROI) (Figure.1B). **(Classic spatially aware AE with attention mechanism)**. Here we introduced the attention mechanisms by assigning to a sample a linear combination of specified inputs. We emphasize attention to histological ROIs (i.e. cell populations of interest like tumor and stroma) and relative distances (Figure.1C-D). **Type of connections:** For fully connected AE all layers are full connected and input features equally weighted. For sparse connections, we introduce sparsity by enriching the model with more biological sense, thus "regrouping" features that have biological relationships(Alessandri et al., 2020). These relationships are based on expert cell annotations. In practice, this is done by enforcing sparse connections in the first layer of the encoder. Hidden nodes of the first layer only receive inputs from the nodes associated to this node (Figure.1B). **Hyperparameter tunning and clustering:** Model hyperparameters (i.e. number of layers, dimension of the hidden layers, dimension of the latent space, batch size, Wasserstein regularization weights, learning rate, activations functions) were optimised using a grid-search paradigm (only for the spatial attention AE). A trained model for each combination (model, connections type) was used to compute the automated mutual information (AMI) out of different clustering strategies: Kmeans, Phenograph(Levine et al., 2015) and Spatialsort(Lee et al., 2022).

| Loss metrics | Model | Connections | Train | Test 1 | Test 2 |
|---|---|---|---|---|---|
| Reconstrution error (MSE) | Classic | Fully | $1.31 \pm 0.04$ | $2.13 \pm 0.06$ | $2.01 \pm 0.05$ |
| | | Sparse | $1.67 \pm 0.09$ | $2.48 \pm 0.11$ | $2.37 \pm 0.10$ |
| | Classic Spatial | Fully | $1.48 \pm 0.04$ | $2.58 \pm 0.07$ | $2.38 \pm 0.07$ |
| | | Sparse | $1.84 \pm 0.08$ | $2.99 \pm 0.14$ | $2.76 \pm 0.12$ |
| | Spatial Attention | Fully | $1.44 \pm 0.05$ | $2.28 \pm 0.06$ | $2.17 \pm 0.06$ |
| | | Sparse | $1.84 \pm 0.10$ | $2.65 \pm 0.11$ | $2.55 \pm 0.11$ |

Table 1: Reconstruction error and comparison between different autoencoders.

## 3. Results

The results (Table 1) were smoothed with 50 iterations per model and the sliced-Wasserstein distance was evaluated 50 times for each embedding. The values are presented as mean $\pm$ std. The autoencoder succeeds in capturing important features of the data. The original data is indeed well reconstructed from embeddings of dimension 16 (versus an original dimension of 43). The regularization using Wasserstein distance enables the latent space to have the shape of a Gaussian distribution, which is beneficial for the clustering techniques (Figure.1E). Sparse connections and spatial information impact is however harder to notice. Hard to analyse the results concerning clustering as there is no ground truth novels phenotypes.

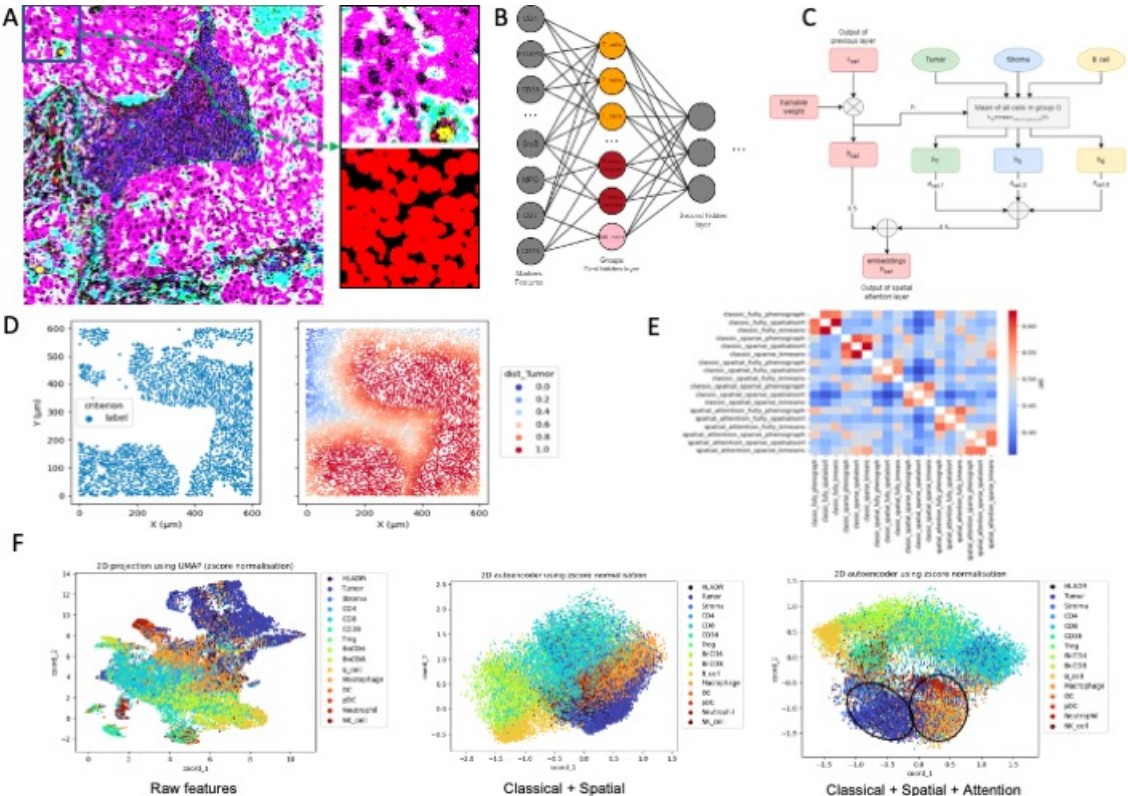

Figure 1: Latent representation. A) Illustration of selected channels from raw IMC as well as high power field and mask of selected region. B) Design of sparse connections guided by biological rationale. C) Spatial attention mechanism using major histological regions as anchors. D) Illustration of a ground truth histological ROIs. E) Clustering AMI criteria based on different AE. F) 2D projections of cells (sampled from the training dataset I) using normalised raw features (left) and AE embeddings with spatial information integrated (middle) as well as attention mechanism (right).

## 4. Conclusions

Our experiments shows that AE succeeds in capturing important features of the data however the impact of spatial information it is harder to notice using AMI or the Wasserstein distance and by selecting a specific region as well as other evaluation metric. The influence of the model and the type of connections (sparse or full) can clearly observed when looking at the overlay of cell phenotypes labels on AE embeddings (Figure.1F). The different cell populations are more well defined with AE (middle and right graphics). The overlapping cell phenotypes (indicated with black oval shapes in Figure.1F) within tumour we can identify a cluster of cells with both DC and tumour cell inside recently shown to highlights a patient not-likely to respond to immune therapies(Cohen et al., 2022)(Oh et al., 2020).

## Acknowledgments

Acknowledgments withheld.

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
