# OpenReview forum: "Applying spatial attention-based autoencoder learning of latent representation for unsupervised characterization of tumor microenvironment"
_MIDL.io/2023/Short_Paper_Track — MIDL 2023 Short paper track Poster_

### Official Review · Reviewer_bHVY · 2023-04-10
**Identify tumor microenvironment**

**Rating:** 6
**Confidence:** 5

**Review:**

This paper combines information on cellular phenotypes with the physical proximity of cells to identify organ-specific microanatomical in the tumor microenvironment
The advantages of the paper include:
+ This is an essential clinical problem for cancer pathology
+ The downstream cell spatial analysis is very interesting for the community
The limitation of the paper includes:
- The quality (resolution) of the images seems low, which makes them difficult to interpret
- The performance is marginally improved compared with baseline methods

---

### Official Review · Reviewer_s7wz · 2023-04-20
**Clarity of contribution, methodology, and results can be improved**

**Rating:** 5
**Confidence:** 3

**Review:**

Summary
The paper proposed an autoencoder framework with a spatial attention module to learn representations of the tumor microenvironment. The paper conducted experiments with several baseline comparisons.

Strengths
The motivation of leveraging spatial relationships and representations for learning and analyzing cell structure phenotyping looks promising.

Weaknesses
(1) The contribution and novelty of the proposed method are not clear. As the authors introduced, there were already some works applying attention mechanisms in biological data. The contribution of the proposed method compared with existing works is thus not clear.
(2) The proposed method is difficult to follow. First, the descriptions of the methods are text only. Without necessary formulation and equations, it is very hard for readers to understand what is "introduced", "is used", and "this is done by", etc. Second, the diagrams of the proposed method in Figure 1 are unreadable due to the blur and small text.
(3) As shown in Table 2, does the proposed method (Spatial Attention) perform worse than baselines (Spatial Attention's MSE is larger)?